# Virulence Profiles of Wild-Type, P.1 and Delta SARS-CoV-2 Variants in K18-hACE2 Transgenic Mice

**DOI:** 10.3390/v15040999

**Published:** 2023-04-19

**Authors:** Yasmin da Silva Santos, Thais Helena Martins Gamon, Marcela Santiago Pacheco de Azevedo, Bruna Larotonda Telezynski, Edmarcia Elisa de Souza, Danielle Bruna Leal de Oliveira, Jamille Gregório Dombrowski, Livia Rosa-Fernandes, Giuseppe Palmisano, Leonardo José de Moura Carvalho, Maria Cecília Rui Luvizotto, Carsten Wrenger, Dimas Tadeu Covas, Rui Curi, Claudio Romero Farias Marinho, Edison Luiz Durigon, Sabrina Epiphanio

**Affiliations:** 1Laboratory of Cellular and Molecular Immunopathology of Malaria, Department of Clinical and Toxicological Analysis, Faculty of Pharmaceutical Sciences, University of São Paulo, São Paulo 05508-000, Brazilsabrinae@usp.br (S.E.); 2Laboratory of Malaria Research, Oswaldo Cruz Institute, Oswaldo Cruz Foundation, Rio de Janeiro 21040-900, Brazil; 3Laboratory of Clinical and Molecular Virology, Department of Microbiology, Institute of Biomedical Sciences, University of São Paulo, São Paulo 05508-000, Brazil; 4Laboratory of Experimental Immunoparasitology, Department of Parasitology, Institute of Biomedical Sciences, University of São Paulo, São Paulo 05508-000, Brazilmarinho@usp.br (C.R.F.M.); 5Unit for Drug Discovery, Department of Parasitology, Institute of Biomedical Sciences, University of São Paulo, São Paulo 05508-000, Brazil; 6Hospital Israelita Albert Einstein, São Paulo 05652-900, Brazil; 7GlycoProteomics Laboratory, Department of Parasitology, ICB, University of São Paulo, São Paulo 05508-000, Brazil; 8School of Natural Sciences, Macquarie University, Sydney 2109, Australia; 9School of Veterinary Medicine of Araçatuba, São Paulo State University, São Paulo 16050-680, Brazil; 10Butantan Institute, São Paulo 05508-040, Brazil; 11Ribeirão Preto Medical School, University of São Paulo, Ribeirão Preto 14049-900, Brazil; 12Interdisciplinary Program of Health Sciences, Cruzeiro do Sul University, São Paulo 08060-070, Brazil; 13Immunobiological Production Section, Bioindustrial Center, Butantan Institute, São Paulo 05503-900, Brazil; 14Scientific Plataform Pasteur/USP, University of São Paulo, São Paulo 05508-020, Brazil

**Keywords:** SARS-CoV-2, COVID-19, Gamma variant, Delta variant, variants of concern, K18-hACE2

## Abstract

Since December 2019, the world has been experiencing the COVID-19 pandemic caused by the severe acute respiratory syndrome coronavirus 2 (SARS-CoV-2), and we now face the emergence of several variants. We aimed to assess the differences between the wild-type (Wt) (Wuhan) strain and the P.1 (Gamma) and Delta variants using infected K18-hACE2 mice. The clinical manifestations, behavior, virus load, pulmonary capacity, and histopathological alterations were analyzed. The P.1-infected mice showed weight loss and more severe clinical manifestations of COVID-19 than the Wt and Delta-infected mice. The respiratory capacity was reduced in the P.1-infected mice compared to the other groups. Pulmonary histological findings demonstrated that a more aggressive disease was generated by the P.1 and Delta variants compared to the Wt strain of the virus. The quantification of the SARS-CoV-2 viral copies varied greatly among the infected mice although it was higher in P.1-infected mice on the day of death. Our data revealed that K18-hACE2 mice infected with the P.1 variant develop a more severe infectious disease than those infected with the other variants, despite the significant heterogeneity among the mice.

## 1. Introduction

Coronavirus disease 2019 (COVID-19), caused by the RNA virus SARS-CoV-2 (severe acute respiratory syndrome coronavirus 2), was declared a pandemic in March 2020 and has accounted to date for more than 6.8 million deaths worldwide [1]. The clinical course of COVID-19 varies from asymptomatic to severe illness, with manifestations including cough, sputum, shortness of breath, fever, muscle, and joint pain, headache, fatigue, and digestive symptoms [2,3]. Lungs in patients appear congested, presenting important broncho-alveolar inflammation and intracytoplasmic viral inclusion bodies. Additional findings include intra-alveolar hemorrhages, fibrinoid necrosis, and intra-alveolar neutrophil infiltration with substantial inflammatory edema [4]. Patients with severe manifestations of the disease present diffuse alveolar damage (DAD), and recent findings highlight the presence of pulmonary vascular endothelialitis, thrombosis, and angiogenesis [5]. A histopathological evaluation of the transgenic K18-hACE2 mouse model used to study the SARS-CoV-2 infection, revealed similar tissue dysfunctions, with lungs showing a mixed inflammatory infiltrate (macrophages and neutrophils), consolidation, and occasional intra-alveolar and perivascular hemorrhages. Interstitial pneumonia associated with alveolar histiocytosis can also be observed, as well as mild type II pneumocyte and endothelial hyperplasia, vasculitis, and edema [6,7].

Since its emergence, the virus’s genetic evolution has been intensely assessed in the knowledge that viruses mutate, leading to new variants [8,9,10]. This effort is extremely important to investigate whether the current diagnostic tests remain adequate, to evaluate the clinical severity of upcoming variants, to understand how the virus spreads, and to measure the efficacy of the vaccines. Multiple genetic lineages of SARS-CoV-2 have been reported globally, prompting the characterization of variants of interest (VOIs) and variants of concern (VOCs). According to the World Health Organization (WHO), mutations in the VOCs’ spike (S) proteins, or other genes of biological importance, effectuate a possible increase in the transmission of the virus and in its capacity to escape from the neutralizing antibodies generated by the main vaccines applied worldwide, posing an additional threat to public health. To date, the Alpha (B.1.1.7), Beta (B.1.351), Gamma (P.1), and Delta (B.1.617.2 and AY) lineages have been included in the VOC group to ensure their surveillance around the globe [9,11,12,13,14,15,16,17,18].

These SARS-CoV-2 variants are associated with an increased risk of reinfection [19] and appear more transmissible than the wild-type (Wt) SARS-CoV-2, isolated primarily in Wuhan at the beginning of the pandemic. However, it is not clear whether there are differences among these variants concerning the severity of the illness [12,20].

In early December 2020, the P.1 variant was identified in Manaus (AM, Brazil), causing widespread infection despite the high seroprevalence against the Wt SARS-CoV-2 strain observed in the city [21]. This variant carries 17 amino acid changes, ten of which are located in its S protein, including three assigned as alarming (N501Y, E484K, and K417T) [22]. Other analyses indicate that these mutations increase the probability of disease transmission [23,24,25,26,27,28,29]. Interestingly, Gamma dissemination has been linked to a higher number of young patients developing severe complications [30]. The Delta strain was first isolated in India in December 2020 and has been considered to spread more quickly, causing more severe cases than the other variants. It was noted that with this lineage the effectiveness of the vaccines was reduced, leading to reinfections/infections and symptomatic illness. However, as reported by the WHO, vaccination still prevents severe illness, hospitalizations, and deaths caused by the Delta strain of SARS-CoV-2 [31,32]. Although the pandemic has lessened in severity, with the circulation of the omicron subvariants (BQ.1 and XBB) and their reduced pathogenicity [33], our research explored and characterized this life-threatening illness, comparing the aggressive variants, such as the Wt strain and the P.1 and Delta variants. A study of the variants that cause the more severe disease is of fundamental importance considering that new, more virulent variants could emerge and re-escalate the pandemic [34,35].

## 2. Materials and Methods

### 2.1. Animal Ethics

Female or male 6- to 8-week-old K18-hACE2 [36] transgenic mice expressing the human ACE2 receptor (B6.Cg-Tg (K18-ACE2)2Prlmn/J—The Jackson laboratory-strain #034860) were initially imported from The Jackson Laboratory (Sacramento, CA, USA). Colonies were kept in the animal facility of the Department of Parasitology, Institute of Biomedical Sciences, University of São Paulo (ICB-USP).

All mice used in this study were managed according to the practices defined and approved by the animal welfare committee (CEUA n° 3858020621) from IBS-USP. Moreover, the experiments followed the ethical guidelines of CONCEA (National Council for Control of Animal Experimentation) according to Brazilian Federal Law n° 11. Mice were provided with ad libitum diets and kept on ventilated shelves under controlled temperature, humidity, and lighting conditions.

The K18-hACE2 transgenic mice were euthanized by intramuscular injection of ketamine (150 mg/kg) and xylazine (15 mg/kg) on the 5th, 6th, or 7th day post-infection (dpi). All efforts were performed to minimize animal sorrow. Signs of suffering or imminent death were criteria for euthanasia (humane endpoint).

### 2.2. Infection with the Wt SARS-CoV-2, P.1, and Delta variants and the Experimental Groups

The infection experiments were carried out at a Biosecurity Level 3 laboratory (BSL3) from the Department of Parasitology (ICB-USP), following the code of good practice and guidelines established for a Biosecurity Level 3 laboratory. The K18-hACE2 mice were infected intranasally with 1 × 10^5^ PFU/mouse of Wt strain or the P.1 or Delta SARS-CoV-2 variants, kindly provided by Professor Edison Luiz Durigon (ICB-USP, Laboratory of Clinical and Molecular Virology, São Paulo, Brazil). The viruses obtained from the nasopharyngeal swabs were isolated in Vero E6 cells. The initial passages and virus titrations were also performed using Vero E6 cells, as well as CCL-81 containing the Gibco™ Cell Culture DMEM medium (Thermo Fisher Scientific, Waltham, MA, USA), according to the protocol described by Bastos et al. (2020) [37]. All viral cultures used in this work came from the 3rd cell passage. The concentration of the viral culture (Wt:1.44 × 10^7^ PFU; P.1: 6.67 × 10^6^ PFU; Delta: 3.09 × 10^5^ PFU) was diluted in PBS 1X to reach a concentration of 1 × 10^5^ PFU/30 μL/mouse. Virus inoculations were performed under isoflurane anesthesia. The female or male K18-hACE2 mice were divided into four groups: the Control and three different groups infected with the Wt strain, the P.1 variant, or the Delta variant. Control group animals received PBS 1x intranasally.

### 2.3. The Clinical Evaluation of the SARS-CoV-2 Infection

The clinical manifestations of the disease caused by the Wt strain, or the P.1, or Delta variants, were evaluated according to previously published procedures [16,17,38,39]. Mice were weighed before infection on day 0 and evaluated daily for body weight loss, and other clinical signs from day one to seven. The clinical manifestations analyzed were posture, breathing, coat, spontaneous behavior, and eye closure. Each parameter ranged from 1 to 5, accordingly to the intensity of these clinical manifestations: 1 = no clinical manifestation, 2 = mild, 3 = moderate, 4 = moderate to severe, and 5 = severe clinical manifestation (implementation of the humane endpoint). The severity of the disease was the sum of the daily clinical scores together with the weight loss. The experimental design aimed to follow the infection for seven days. However, the humane endpoint was reached earlier in some cases and mice were euthanized from the 5th dpi.

### 2.4. Whole-Body Plethysmography Evaluation

To understand the influence of the viral infection on respiratory function, whole-body plethysmography (WBP; DSI Buxco Respiratory Solutions, DSI Inc., Goodfield, IL, USA) was used. This apparatus, suitable for longitudinal studies, is non-invasive, with a chamber equipped to measure pressure, flow, and volume changes in the respiratory function of individual mice. Without anesthesia or restraint, the animals were placed in plethysmographic chambers for ten minutes on the 3rd, 5th, and 7th dpi. The Fine Point software recorded the parameters of respiratory frequency (RF)—evaluating the number of breaths per minute (bpm), enhanced pause (Penh)—calculated by the ratio between the respiratory pause in relation to the inspired and expired volume inside the chamber, and the exhalatory flow curve (Rpef)—the ratio between the time to peak expiratory flow and the total expiratory time [40,41].

### 2.5. Quantification of the Viral Load by RT-qPCR

For this analysis, we used samples from lung and oral swabs from the control and the three groups of infected mice collected on the day of death (time point or endpoint). The animal was restrained with its head positioned so that its mouth was ajar and the swab (Crystal Monofilament Rod Swab ryon fiber (110 mm; Batch 010321107-1; Swab Brazil Industry of Laboratory Products Ltd.) was inserted and gently rotated. The samples were stored in a VTM (viral transport medium) at −80 °C until use. The oral swabs and lung samples were macerated using the MagNA Lyser equipment (Roche Diagnostics, Mannheim, Germany) and subsequently centrifuged at maximum speed for 15 min. A 150 µL aliquot of the supernatant was mixed with 200 µL of Lysis Buffer (BioMerieux, Lyon, France). The swabs were shaken in a vortex mixer and subjected to the same process. Subsequently, the extraction of total nucleic acid was performed according to the manufacturer’s protocol (Nuclisens MagMax, BioMerieux, Lyon, France).

With the genetic material already extracted, the samples were submitted for diagnosis by RT-qPCR to detect SARS-CoV-2, and the mammalian ribonucleoprotein (RNP) was used as an extraction control. These procedures utilized the AgPath-ID™ One-Step RT-PCR Kit (Applied Biosystems/Life Technologies, Austin, TX, USA) and the 7500 Real Time System equipment (Applied Biosystems/Life Technologies, Foster City, CA, USA) with protocols adapted from Corman et al. (2020) [42]. Both forward and reverse primers and probes were used in equal proportions at a concentration of 10µM for the detection of SARS-CoV-2 (FAM) and RNP (VIC). As negative and positive controls, in all reactions, we used nuclease-free water and a clinical isolate in the Vero-E6 cell culture, duly tested and standardized.

The cycles used were: 45 °C for 15 min, once; 95 °C for 10 min, once; 95 °C for 15 s and 57 °C for 1 min, 45 times. For the conclusion of the diagnosis, it was determined that samples with Ct values <37.99 were positive and those with Ct values >42 were negative. Samples with a value between 38 and 41.99 were considered inconclusive and were repeated from the stage of the extraction of the total nucleic acid. Finally, after the detection of the virus by RT-qPCR, we quantified the viral load of the samples in the number of copies/mL. The results revealed a variation between 4.65 × 10^2^ and 4.41 × 10^9^ in the material.

### 2.6. Histopathology Analysis

A fragment of the tissue was fixed in neutral buffered 10% formalin for histopathological analysis. For dehydration, after 24 h, samples were washed with 70% ethanol, and following fixation, they were processed for paraffin embedding to form a solid block. Subsequently, the blocks were cut into 5-µm-thick sections and stained with hematoxylin and eosin (H&E). Histological scores from 0 to 4 for edema, hemorrhage, inflammatory infiltrates, congestion, hemosiderin, fibrin, alveolar emphysema, thickness, atelectasis, vasculitis, diapedesis, necrosis/consolidation, reactive alveolar macrophages, pneumonia, and thrombosis were included in the analysis, where 0 = no alteration, 1 = slight alteration, 2 = moderate, 3 = important, 4 = severe alteration. The severity of the disease was the sum of histopathological lesion scores.

### 2.7. Statistical Analysis

Using GraphPad Prism 8.0 (GraphPad Software, San Diego, CA, USA) or STATA 17.0 software, the data were initially analyzed for normality by the Kolmogorov–Smirnov, D’Agostino–Pearson, and Shapiro–Wilk tests, and Bartlett’s test for variance. Non-parametric variables were compared using the Mann–Whitney rank-sum test, the Kruskal–Wallis test followed by Dunn’s post hoc test, or the Brown–Forsythe and Welch ANOVA tests with the Games–Howell multiple comparisons test. The one-way ANOVA with the Bonferroni multiple comparisons test, or Multiple T-test, were used for parametric variables. For the survival curves, the Log-rank (Mantel–Cox) test was applied.

## 3. Results

### 3.1. The P.1 Variant Induces a More Severe Disease Than the Wt Strain and Delta Variant in ACE2-Transgenic Mice

Animals received intranasal infections of the Wt, P.1, or Delta variants at a concentration of 1 × 10^5^ PFU/mL, as per the literature [43]. On day 5 post-infection, all three groups of infected mice showed losses in weight when compared to the non-infected control group (Figure 1A). Interestingly, at the 5th dpi, no weight loss difference was observed between the Wt and Delta variants compared to the control, and only the P.1 group showed significant weight losses compared to the control (*p* = 0.0034) and the Delta variant (*p* < 0.05) (Figure 1B). These data indicate that the disease generated by the P.1 variant proves to be more severe than the Wt strain or Delta variant (Figure 1A,B).

No gender-based differences were observed (Appendix A), which is similar to Winkler et al. [44].

Furthermore, we determined a clinical score to assess the severity of disease in mice infected with the different strains of the SARS-CoV-2 virus. In addition to body weight loss, we assessed the general condition of the animal, such as posture, breathing, coat, spontaneous behavior, and eye closure, in a score ranging from 1 to 5, where 1 was the absence of signs and 5 was a severe manifestation of ill-health in the aforementioned areas. The P.1-infected mice had the highest (worst) clinical score (Figure 1C and Appendix A) compared to the control group (*p* < 0.0001), Delta group (*p* < 0.05), and Wt group (*p* < 0.05). As 90% of the P.1-infected mice reached a score of 5 at the 5th dpi the humane endpoint was implemented, and they were euthanized. Clinical signs of the disease started to appear within the 2nd dpi for all infected groups.

The animals became clearly lethargic, presenting breathing issues, with eyes slightly closed. The total clinical scores obtained for the Delta variant and Wt strain were similar, as no statistical variations were observed.

In conformity with the data on weight loss and clinical manifestation, the virulent P.1 variant led to severe illness, and mice had to be euthanized on 5th dpi (humane endpoint), while the Wt-mice were euthanized on the 6th and 7th dpi (humane endpoint and timepoint). The Wt-mice mortality was lower at 12.5%, and quite surprisingly, no deaths were observed in the Delta-infected mice euthanized on the 7th dpi (timepoint) (Figure 1D). It is important to highlight that the Wt-infected mice presented a more variable disease, showing some animals as very sick and others with a very light disease behaving similarly to the control group (Figure 1C).

### 3.2. The Whole-Body Plethysmography Demonstrates That the SARS-CoV-2 P.1 Variant Induces More Severe Lung Dysfunction

Pulmonary physiological measurements were performed in non-infected or infected animals using an unrestrained chambered whole-body plethysmograph with measurements of the static lung volume, respiratory frequency (RF), enhance pause (Penh), and exhalatory flow curve (Rpef) [40,41]. On the 3rd dpi, no differences in any parameters were noted across all experimental groups (Figure 2A,C). A substantial decline in RF, however, was observed in the P.1-infected mice on the 5th dpi when compared to the control group (*p* < 0.0001) or Delta-infected mice (*p* < 0.0001).

The heterogeneity of the disease presented by the Wt strain influenced the assessment of lung capacity, also observed in the Rpef. Despite the evident respiratory capacity reduction observed in Wt-infected mice, which was mainly respiratory frequency on the 7th dpi (Figure 2A), no statistical difference was observed compared to the other groups (control or variants) [45]. It is important to note a difference in RF between the Delta- and P.1-infected mice, suggesting that infection with the most recent variant is more severe than the earlier strains or variants. In line with the previous results, the Delta- and Wt-infected mice showed similar RF kinetics on the 5th dpi. Substantial alterations on the Penh were also observed for Wt-infected mice on the 5th dpi, although the comparison between groups was not considered significant. Similar values for Penh were observed in Delta- and Wt-infected mice (Figure 2B). On the 5th dpi, analyses of Rpef showed significance for the P.1- infected mice, when compared with the control, Delta- and Wt- infected mice (*p* < 0.001) (Figure 2C). Although the Wt- or Delta-infected mice showed deterioration of lung function compared to control mice, mean changes were not significant due to a wide variation on the 7th dpi. Again, no differences were observed in animals infected with the Delta variant or Wt strain. Similarly, no statistical differences were observed on the 7th dpi across all experimental groups for the Penh parameter, even though values were increased in the infected groups compared to the control group, demonstrating important lung alterations provoked by SARS-CoV-2 infection (Figure 2B). The P.1 variant-infected mice reached the highest Penh values and the lowest values of Rpef on the 5th dpi, and they had to be euthanized (therefore, data was not collected on the 7th dpi), denoting the worst respiratory capacity of this group compared to the others.

### 3.3. Histopathological Changes in the Lungs of K18-hACE2 Mice after Infection with the Wt Strain or the P.1 and Delta Variants

We next sought to better describe the histopathological changes in the lungs of SARS-CoV-2 infected animals. All infected mice suffered significant pulmonary lesions, but the parameters analyzed varied in intensity (scores 0 to 4). For all parameters (sum) analyzed, the P.1-infected mice showed a higher total histopathological score than the Wt-infected mice (*p* < 0.01), and scores tended to be higher than the Delta-infected mice, although statical significance was not achieved (Figure 3A). Inflammatory infiltrates (Figure 3B,C) were a more evident and frequent finding in 100% (45/45) of infected mice, demonstrated by interstitial pneumonia with thickening of alveolar septa (Figure 3D,E) due to a mix of inflammatory infiltrates (presence of polymorphonuclear and mononuclear cells), reactive alveolar macrophages, and diapedesis. However, there were no significant differences in the inflammatory processes among these three variants, despite the interstitial pneumonia, bronchopneumonia, and necrotizing pneumonia, identified by areas of necrosis and consolidation (Figure 3F,G) in 26.67% (12/45) of lung samples from the infected mice. In addition, vasculitis was observed in 15.55% (7/45) of both venous and arterial vessels (Figure 3H,I). Regarding hemodynamic disorders, congestion was observed in 64.44% (29/45), being more critical in the P.1-infected mice, especially compared to the Wt-infected mice (*p* < 0.05) (Figure 3J,K). Hemorrhages detected in 68.88% (31/45) were seen in the alveolar, peribronchiolar, intrabronchial, perivascular, and subpleural tissues (Figure 3L,M), with hemosiderin in 6.67% (3/45) of cases (Figure 3N,O) or without, suggesting more recent hemorrhages. Interestingly, the P.1-infected mice showed more severe hemorrhages than the Wt- and Delta-infected mice (*p* < 0.05). In addition, thrombosis (Figure 3R,S) was observed in a few cases 6.67% (3/45). Furthermore, although alveolar edema is a critical finding in acute respiratory syndromes, it was observed in a few analyzed fragments of lung tissue 8.89% (4/45) (Figure 3P,Q). Alveolar emphysema (Figure 3T,U) was also observed in 62.22% (28/45) of infected mice. Areas suggestive of fibrin (66.67%; 29/45) were explored and quantified (Figure 3V,X). Atelectasis was also observed in 53.34% (24/45) of mice infected by the three variants. In addition, viral inclusion particles and hyaline membrane formation in the lung tissue of just one of the Delta-infected mice were observed (data not shown).

### 3.4. Quantification of SARS-CoV-2 in the Lungs and Oral Swabs of K18-hACE2 Mice Infected with the SARS-CoV-2 Wt, P.1, and Delta Lineages

Oral swabs and lung tissues from infected and uninfected mice were analyzed by the RT-PCR method for SARS-CoV-2 virus quantification. The amount of virus varied greatly among mice within the same and experimental groups. SARS-CoV-2 virus amplification could be detected on the 5th, 6th, and 7th dpi (day of death: humanized endpoint or time point) in oral swabs and lungs of the infected animals (Figure 4A,C). Differences between the P.1 and Delta variants were evident in the material analyzed; the lungs of both groups had a higher viral load on the day of death, as seen in the results found in the oral swab (*p* < 0.05) (Figure 4A,C). No statistical differences could be observed in the viral load found between oral swabs and lungs from the Wt-infected mice (Figure 4B). On the other hand, comparing the pulmonary viral load on the day of death, we observed that the P.1-infected mice had larger loads than the Delta- and Wt-infected mice. In addition, the viral quantification from the oral swabs was greater in the P.1-infected mice than in the Delta-infected mice (Appendix A).

## 4. Discussion

The SARS-CoV-2 P.1 or Delta variants were associated with increased case fatality [10,12,20]. Several rodent models have been developed to study the pathogenesis of severe COVID-19, especially with K18-hACE2 [8,36,46,47,48,49,50]. Investigating the virulence and pathogenicity of SARS-CoV-2 variants in well-established models is relevant to predicting the lethality of the variants, thus providing public health services with helpful information for decision making and understanding the mechanisms that lead to the increase in the virulence, as with the variants, permitting further exploration of these microorganisms [51,52,53]. K18-hACE2 transgenic mice have been shown to be highly susceptible to SARS-CoV-2, as infection with the virus leads to severe interstitial pneumonia and pathological changes similar to those seen in patients with COVID-19. Usually, the death of infected animals occurs on the 7th dpi, dependent on the inoculum concentration [54,55].

Firstly, our work suggests that the P.1 variant of SARS-CoV-2 causes a more severe disease and a more significant weight loss in infected mice compared to control and Delta-infected mice, but not compared to Wt-infected mice. Liu et al. also observed no difference in this parameter, throughout the infection, between the Delta variant and the Wuhan strain in infected K18-hACE2 mice [47]. Furthermore, Stolp et al. observed very intense weight reductions (almost 20%) at the 5th dpi in Gamma-infected mice. However, at the same time, there were no differences regarding weight loss compared with EU-1, Alpha, and Beta-infected mice [50].

Observing the COVID-19 disease in K18-hACE2 mice, we noted that the main clinical manifestations were more pronounced in P.1-infected mice than in the other experimental groups. Furthermore, the total clinical scores, associated with reduction in weight loss, from the P.1-infected mice showed the worst clinical status. The results showed that the survival rate probability of Delta- or Wt- infected mice was similar. However, the P.1- infected mice, in agreement with the results discussed above, had a worse survival rate, reaching the humanitarian endpoint (on the 5th dpi) before the stipulated experimental endpoint (on the 7th dpi) (Figure 1F), as observed by Stolp et al. [50].

The whole-body plethysmography evaluating lung function confirmed a more severe disease in P.1-infected mice, as both the RF and Rpef parameters were significantly reduced, while some mice showed that the Penh was significantly increased, demonstrating that the respiratory capacity of these animals was compromised (Figure 2). The lung capacity of the P.1 group was the worst recorded, corroborating the fact that the animals reached the humanitarian end point before the time determined for this study, proving the aggressiveness of the mutations present in the Gamma variant. This is the first report of a reduction in pulmonary function measured by the plethysmography chamber in P.1-infected K18-hACE mice. On other hand, using the Wt strain from Milan, Italy (GenBank: MT748758.1), Fumagalli et al. observed the same respiratory rate decrease, reinforcing our data [56]. Our results corroborate the epidemiological data in places where P.1 has spread.

Notably, Freitas et al. suggested a worse pathogenicity and virulence from the P.1 variant, compared to the non-Gamma strains, which increased the mortality of younger patients [57]. In Manaus (Brazil), the P.1 variant caused an extremely serious public health crisis, with lack of available oxygen treatment and high mortality due to the high number of infected people [58]. Faria et al. demonstrated that the P.1 lineage is more transmissible and lethal than the others, as it carries 17 important mutations, including three in the virus’s S protein (K417T, E484K, and N501Y) [24]. The storage of mutations in the Gamma variant is responsible for the greater affinity for hACE2 and its infectious potential [59]. In fact, accumulated mutations can affect the disease transmission. However, a higher mutation rate will not necessarily lead to more severe disease. For example, the Delta strain exhibits 10 mutations in the S protein (T19R, G142D, 156del, 157del, R158G, L452R, T478K, D614G, P681R, and D950N) and has proved to be less aggressive than the P.1 strain [60]. On the other hand, the omicron variant, described in late 2021 in Africa, has more than 30 mutations in addition to deletions in the gene. These mutations promote greater transmissibility, as well as a greater avoidance of the host antibodies.

Our data demonstrate that Delta-infected mice had a similar disease severity compared to the Wt strain when considering weight loss, manifestation of clinical signs, and lung function (Figure 1 and Figure 2). Liu et al. demonstrated similar clinical profiles, and SARS-CoV-2 quantification on the 6th dpi between the Wt strain and Delta variant in female K18-hACE2 mice infected with 10^4^ PFU, showing no significant statistics [47]. Contrary to what we demonstrated, Hu and colleagues [61] suggested that the Delta variant can cause severe disease in infected humans compared to the Wt strain of SARS-CoV-2. Another recent report found that Delta-infected patients had twice the risk of hospitalization than those with Alpha [62], which can be explained by the histopathological changes similar to the P.1 variant observed in Figure 3A. However, this variant proved to be more transmissible in humans and was 15 times more excreted in saliva than the Wt strain of SARS-CoV-2, which suggests the need for further research for the SARS-CoV-2 variants associated with mutations that can lead to the development of a more transmissible disease with the possibility of being more virulent, such as the Delta and Gamma strains [63]. The three variants showed distinct and non-uniform morphological patterns, but the pulmonary inflammatory state was consistent and related to the cause of death of the K18-hACE2 mice, as well as hemorrhage, thrombosis and, in some cases, the presence of edema. Similar to the findings of the present study, Golden et al. [64] observed fibrin thrombi and inflammatory infiltrate and collapsed alveolar septa, with lung consolidation, edema and vasculitis. Most of the pulmonary lesions observed in patients who died due to COVID-19 were similar to the work presented with diffuse alveolar damage, mixed inflammatory infiltrate (mono- and polymorphonuclear), edema, intra-alveolar hemorrhages, congested vessels, presence of fibrin and thrombi, syncytial giant cells, type II hyperplasia, and the formation of the hyaline membrane [7]. However, we did not observe giant cells or type II hyperplasia in the lungs of infected mice. Although the loss of alveolar architecture was frequently detected, we only identified the hyaline membrane formation in one Delta-infected mouse. Potentially, should our infected mice have survived for more extended periods, they could present with these histopathological findings. Viral particles in the lungs were also observed in one Delta-infected mouse. It is essential to emphasize the impossibility of determining a definitive pulmonary histomorphological signature in experimental COVID-19. The use of qRT-PCR showed that the viral load varied considerably within groups of infected animals. Due to the large dispersion found in the viral load analyses, no differences were observed between oral and lung swabs from animals infected with the Wt strain at 6/7 dpi, unlike that shown by the Delta variant in the same period. Considering the viral load evaluated in the animals’ oral swabs, the highest number of gene copies was found in the P.1 strain at 5 dpi (Figure 4A). Indeed, it has been suggested that patients with severe COVID-19 tend to have a high viral load and a longer period of virus shedding [65,66]. In line with our results for qRT-PCR, it was estimated that P.1 has an effective reproduction number 2.2 times greater than the parental strain, suggesting that it is 2.5 times more contagious than the original coronavirus [67]. The present work emphatically shows that the Gamma (P.1) strain was, in fact, more virulent and lethal than the others and, interestingly, that the effects of the Delta variant are similar to those of the Wt strain in aspects such as weight loss and changes in clinical manifestations. We observed the impact of P.1 in Brazil with the brutal increase in lethality from February 2021, when it spread, after the initial outbreak in Manaus (Amazonas/Brazil), to the rest of the country. These findings give more solidity, robustness, and relevance to humanized transgenic mice experimental models. Gamma strains share mutations in regions that confer greater infectivity and immune escape. In particular, the N501Y region, providing greater binding strength to the hACE2 receptor and the E484K region, is responsible for the decrease in the response by monoclonal antibodies. The combination of mutations that induce greater infectivity and immunological evasion makes P.1 more aggressive compared to other variants, as demonstrated in our results [68,69].

In conclusion, although the P.1 strain showed more pathogenicity in the majority of infected mice, we emphasize that, regardless of the variant, there are different gradations in clinical manifestations, pulmonary lesions and capacity, viral load, and disease outcomes, denoting the heterogeneity of experimental COVID-19, congruent with patients naturally infected with SARS-CoV-2. A robust model can successfully reproduce human infections, opening the door to studies of the disease mechanisms and the determination of why mutations found in the variants of different lineages impact the pathogenesis and lethality of the disease. This point is crucial for the development of new therapies and vaccines to control and prevent the disease [18,70]. A better characterization of these murine models exposed to the more aggressive virus lineages is critical to understanding the pathogenesis of this fatal disease and testing the effectiveness of the vaccines and treatments against the increasing virulence of the SARS-CoV-2 mutations.

## Figures and Tables

**Figure 1 viruses-15-00999-f001:**
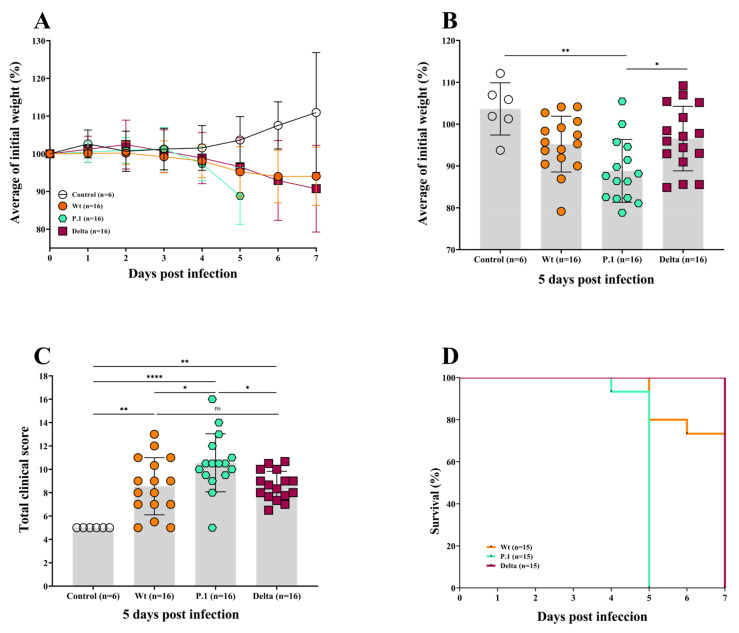
The P.1 variant induced a greater weight loss and clinical manifestation than the Delta variant and Wt strain in ACE2-transgenic mice. (**A**,**B**) The average of weight loss of K18-hACE2 transgenic mice infected intranasally with 10^5^ PFU of the Wt strain and the P.1 and Delta variants. The P.1-infected mice were euthanized on the 5th dpi, and the Wt- and Delta-infected mice on the 7th dpi. Analysis of body weight (**B**) on the 5th dpi showed significant differences between the P.1 variant and the control. The clinical manifestations of the disease in K18-hACE2 mice were more pronounced in the group infected with the P.1 variant when compared to the Delta variant or the Wt strain (**C**). The probability of survival was reduced when the mice were infected with the P.1 variant compared to Delta or Wt strain (**D**). The hypothesis test performed by a one-way ANOVA, with Bonferroni multivariance analysis for the parametric variables, or Brown–Forsythe and Welch ANOVA tests with the Games–Howell multiple comparisons test for values with asymmetric distribution, the survival Log-rank (Mantel–Cox) test was applied, using GraphPad Prism 8.0, *p* < 0.05 was considered significant (* *p* ≤ 0.05; ** *p* ≤ 0.005; **** *p* ≤ 0.0001).

**Figure 2 viruses-15-00999-f002:**
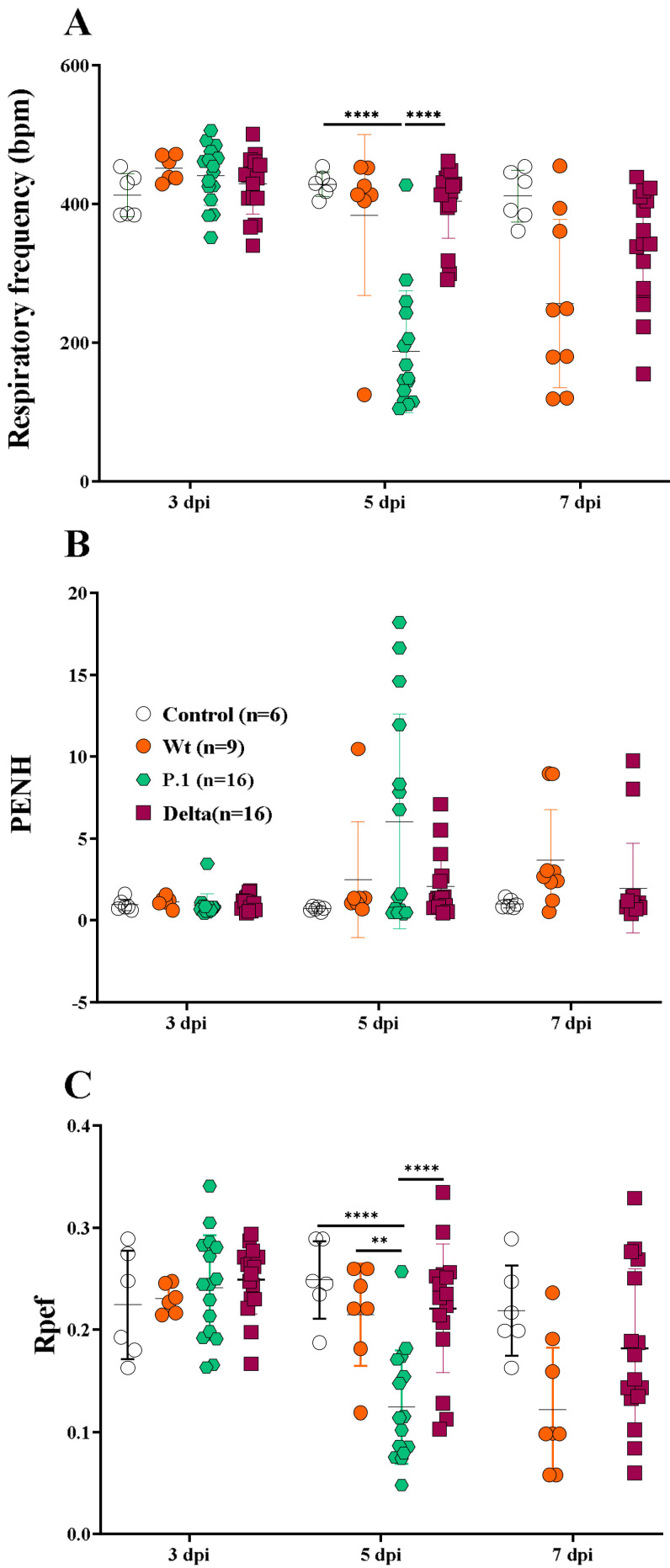
The whole-body plethysmography indicates severe lung dysfunction in the P.1-infected mice. The evaluation of lung function for respiratory frequency (**A**), enhanced pause (Penh) (**B**), and exhalatory flow curve (Rpef) (**C**) in the Wt strain and the P.1 and Delta-infected K18-hACE mice. A multiple *t* test was applied, using GraphPad Prism 8.0; *p* < 0.05, was considered significant (** *p* ≤ 0.005 and **** *p* ≤ 0.0001).

**Figure 3 viruses-15-00999-f003:**
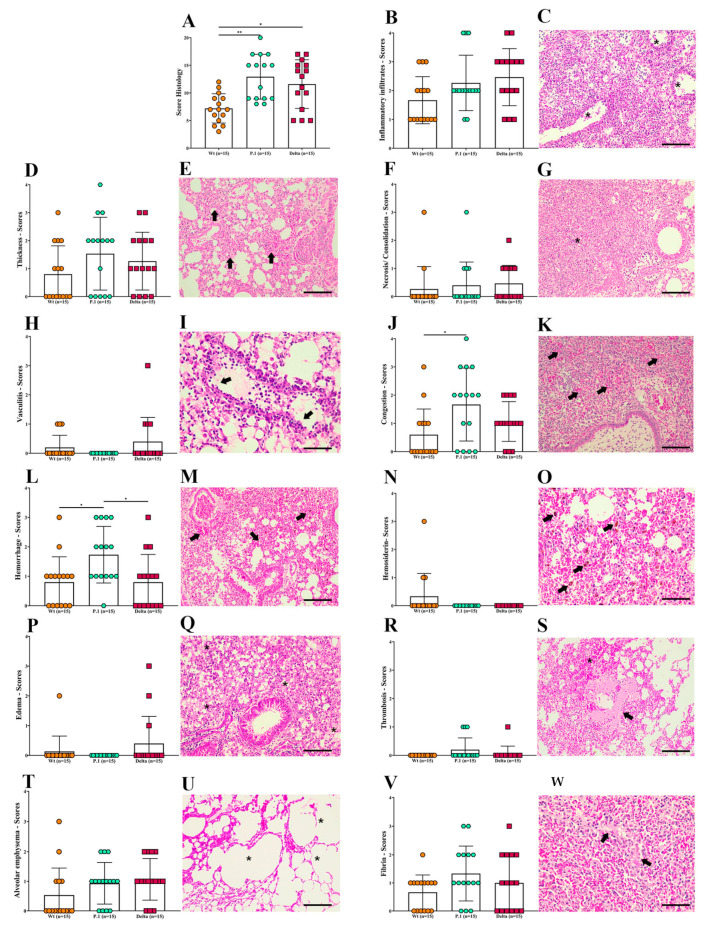
Histopathological changes in the lungs of K18-hACE2 mice after infection with the Wt strain, or the P.1 or Delta SARS-CoV-2 variants. (**A**) The sum of all histological parameters analyzed per group. (**B**,**D**,**F**,**H**,**J**,**L**,**N**,**P**,**R**,**T**,**V**) Individual histological parameters were quantified in three groups of infected mice. (**C**,**E**,**G**,**I**,**K**,**M**,**O**,**Q**,**S**,**U**,**W**) Representative photomicrographs of SARS-CoV-2-infected mice lungs. (**C**) Severe interstitial pneumonia with a mixed inflammatory infiltrate vasculitis and peri-vascular infiltrate (asterisks) from a P.1-infected mouse on the 5th dpi (20×). (**E**) Interstitial pneumonia with thickness of alveolar septa (arrows), from a Wt-infected mouse on the 6th dpi (20×). (**G**) Interstitial pneumonia showing extensive inflammatory infiltrate and consolidation area (asterisk) from a Wt-infected mouse on the 7th dpi (20×). (**I**) Vein vasculitis and diapedesis (arrows) of polymorph and mononuclear cells from a Wt-infected mouse on the 7th dpi (40×). (**K**) The congestion (arrows) and severe bronchopneumonia with extensive inflammatory infiltrate and consolidation, showing exudate inside bronchioles (asterisks) from a P.1-infected mouse on the 5th dpi (20×). (**M**) Extensive and diffuse hemorrhagic areas (arrows) associated with inflammatory infiltrate from a Wt-infected mouse euthanized on the 5th dpi (20×). (**O**) Brownish hemosiderin pigments (arrows) visualized in an extensive hemorrhagic area in a Wt-infected mouse on the 5th dpi (40×). (**Q**) Alveolar oedema, evidenced by light eosinophilic areas (asterisks) and reactive alveolar macrophages (arrows) from a Wt-infected mouse on the 7th dpi (20×). (**U**) Emphysema alveolar areas (asterisks) in a Wt-infected mouse on the 7th dpi (20×). (**S**) White thrombus (arrow) surrounded by bleeding (asterisks) and congested areas. The right upper area shows emphysema in a P.1-infected mouse on the 5th dpi (20×). (**W**) Fibrin (arrow) and congestion areas (asterisks) in a P.1-infected mouse on the 5th dpi. Scale bars represent 40 μm (objective 20×) and 20μm (objective 40×). Wild type = Wt; dpi = days post infection. Data are representative of three independent experiments. Non-parametric variables were compared using Kruskal–Wallis test which was followed by Dunn’s post hoc test using GraphPad Prism 8.0, *p* < 0.05 was considered significant (* *p* ≤ 0.05 and ** *p* ≤ 0.005).

**Figure 4 viruses-15-00999-f004:**
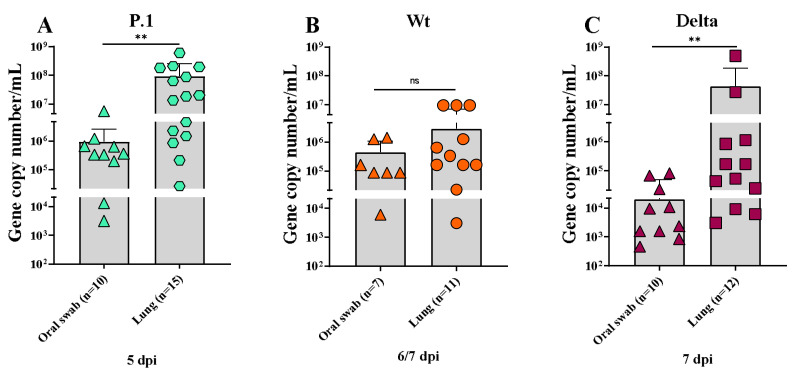
Quantitative detection and viral load analysis in oral swabs and lungs of K18-hACE2 mice. (**A**) Viral load detection by RT-qPCR in oral swabs and lungs of K18-hACE2 mice infected with P.1 variant (**A**), Wt strain (**B**), and Delta variant (**C**) on the day of death (timepoint or endpoint). Test performed by Mann–Whittney test using GraphPad Prism 8.0, considering *p* < 0.05 as significant (** *p* ≤ 0.005).

## Data Availability

The datasets used and/or analyzed during the current study are available from the corresponding author on reasonable request.

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
