# Peer review of "Virulence Profiles of Wild-Type, P.1 and Delta SARS-CoV-2 Variants in K18-hACE2 Transgenic Mice"

_viruses, 2023, doi:10.3390/v15040999_

Round 1

Reviewer 1 Report

In general, the data is clearly described and is relevant for SARS-Cov-2 research, especially for researchers interested to better explore the pathogenesis of the disease caused by emerging SARS-Cov-2’s new variants using the experimental infection model K18-hACE2 mice. 

The experimental data indicates that the P.1 strain is more pathogenic in the experimental infection model K18-hACE2 than the Delta strain and control Wt, which demonstrate the robustness of the experimental infection model.

Although these results are consistent with data in humans, the base of mechanisms for P.1 severity remains unknown. However, considering the limitations of the experimental work performed inside a Biosafety Level 3 (NB-3) working environment, the data already presented is valuable and serve as reference guide for further analysis in the experimental infection model K18-hACE2 .

Minor revisions are recommended to ensure it is fully fit for publication.

Reviewer 2 Report

The study provides comprehensive information on the physiological viral load, lung function, and histopathological analysis of wild-type, P.1, and Delta SARS-CoV-2 variants in K18-hACE2 transgenic mice. The author claimed that K18-hACE2 infected with P.1 variation had a more severe infectious disease than delta and wild-type variant. The generated data and methodologies have been presented by the authors in a reasonable way to support their hypothesis. Before making a final statement, however, the following issues should be aptly addressed. The conclusions of this investigation, in my opinion, merits publishing. Before publishing, a significant revision to the manuscript is required.

Major

1.      Author must ensure that their article is free of grammatical errors and that the abstract includes the correct objective, justification, concise methodology, findings, and conclusion.

2.      In the introductory part, lines 85-87 should be rephrased in a meaningful manner, line 87 requires references, lines 92-95 should be revised, as the phrase is unclear.

3.      In the method section, the whole-body plethysmography evaluation should be described for general readers.

4.      In the result section, Figure1A lacks legends, and Figure3 needs high resolution quality image.

5.      In section 3.4 a detailed description must be included

6.      Limitations and prospective directions should be addressed in the Discussion sections. Please include a conclusion that summarizes your findings in broad terms and gives the reader a taste of what the article is about.

7.      Why P1 is more infective than other variations in terms of mutations is something that the authors should explain and interpret in the discussion section. The research paper will be useful to its readers because there are many sources that may be cited within it. Of the many studies available, I'd like to highlight a few that go into greater depth on this topic: Check these references: PMID: 36679867 PMID: 36195045.

8.      The manuscript has to be reviewed for references throughout because certain references are missing for example line 87-89.

9.      Throughout the discussion, there are a few lines that are quite long, there are too long connectors and linkers in lines 350-352 or 362-366. Please clarify the sentence in a more accurate English form in lines (377-379) and 380-381.

Minor

1.Wild type or wt. should be uniform throughout the manuscript.

2. All abbreviations should be defined at the first mention.

3. English grammar, especially tense usage, and punctuation/full stop usage, should be evaluated   as a whole.

Reviewer 3 Report

Da Silva Santos and colleagues assess the disease course of SARS-CoV-2 VOCs in comparison to non-infected control mice and the original Wuhan strain, in K18-hACE transgenic mice, that are routinely used for this kind of research questions. In line with previous literature they find an accelerated and more severe disease course upon infection with the SARS-CoV-2 P.1 strain. Importantly they extend their research to whole-body Plethysmography and find enhanced and earlier respiratory dysfunction especially with the P.1 VOC.

Major comments:

1.     Similar work has already been performed by other labs that is not referenced in the manuscript. E.g. PMIDs 36037222, 35062016, 35420469, 35602059 and 36849513 for delta virus infection and PMID 35134331 for P.1 virus infection. While the authors do not claim absolute novelty of their findings, they should at least mention and refer to earlier work. In the discussion, they compare their results nicely to observations in human patients, but entirely ignore results obtained from research in K18-hACE2 mice and their results are actually nicely in line with previous infections of mice with P.1. The performed whole-body plethysmography on the other hand has, to the knowledge to this reviewer, not been performed for P.1 infected mice and represents the actual novelty of this paper and should be highlighted.

2.     Clinical score: Figure 1C. The clinical score is only depicted for day 5 post infection and therefore omits plenty of information that must be available to the authors. It would be extremely beneficial if the disease score would be displayed in a time course, as for body weight (Figure 1A) and the reader would extremely benefit if further details on the disease parameters would be provided, such as differences in symptoms in between infections with the different VOCs.

3.     Histological score: Line 287 f. claims that P.1 mice have a higher histopathological score than delta infected mice. This is not shown in Figure 1A, here both P.1 and Delta infected mice show an increased histopathological score as compared to WT infection. Please rephrase accordingly. Therefore the statement that the histology data explains the higher pathogenicity of P.1 is not supported by the data presented.

4.     Line 340 ff., viral gene copy numbers. Here the text explaining the data is entirely missing in the manuscript! In addition, a time course of the viral copy numbers in oral swaps would have been nice, as oral swaps can be performed noninvasively. There is no comparison of viral loads in between the different infections, making conclusion on the impact of viral load on pathogenicity impossible.

Minor points:

1.     Line 117 ff. the authors should mention in which cell lines the virus used for infection was generated and how virus titers were assessed. Why did the authors use PBS for mock infection, but not DMEM, in which the virus was dissolved? Also, passage number of the virus in cell culture would be important, as the spike protein usually quickly adapts to cell culture conditions.

2.     Line 150 ff. the authors describe usage of nasal swaps in their material and methods, but oral swaps in their Figure 4. Please clarify which method is correct and please describe in more detail how these swaps were obtained, e.g. which kind of cotton swaps. M+M also claims that nasal swaps were taken at different time points and a figure of the time course of viral load in the nasal/mouth swaps would have been nice.

3.     Line 211 f. the authors claim that no gender specific differences were observed, but genders were not directly statistically compared in Supplemental Figures S1 and S2, plus differences to the mock infected animals differ in between male and females. Please clarify.

4.     Line 205 claims that there is a significant loss of body weight upon infection shown in Figure 1, but Figure 1 does not include any statistics, therefore please omit the term significant. Furthermore, the text claims that infections with all variants induce a weight loss, but Figure 1B only reaches statistical significance for P.1, please change the phrasing accordingly.

5.     Explain clinical and histological scores better. The text and material and methods says that scores from 1 to 5 or 1 to 4 were given, but Figure 1 C and 3A depict significantly higher scores. While this reviewer understands that is the accumulation of single scores for each parameter, this is nowhere explained in the manuscript. Please specify.  

6.     7 days post infection is the final time point for all infections, but it is not mentioned in the text whether all animals reached euthanization score on day 7, or whether the experiment was finalized by the final organ harvest irrespective of the disease score. This should also be reflected in Figure 1D.

7.     Why were only 6 out of 16 WT virus infected mice assessed by whole-body plethysmography, if data variation was so high?

Respiratory frequency is sometimes abbreviated as FR, sometimes as RF, please unify. Please discuss that RF results are actually in line with the oberservations in Fumagalli et al., 2022, highlighting interesting differences for P.1.
